# Recent Advances and Future Perspectives of In Vivo Targeted Delivery of Genome-Editing Reagents to Germ cells, Embryos, and Fetuses in Mice

**DOI:** 10.3390/cells9040799

**Published:** 2020-03-26

**Authors:** Masahiro Sato, Shuji Takabayashi, Eri Akasaka, Shingo Nakamura

**Affiliations:** 1Section of Gene Expression Regulation, Frontier Science Research Center, Kagoshima University, Kagoshima 890-8544, Japan; stylistics777@yahoo.co.jp; 2Laboratory Animal Facilities & Services, Preeminent Medical Photonics Education & Research Center, Hamamatsu University School of Medicine, 1-20-1 Handayama, Higashi-ku, Hamamatsu, Shizuoka 431-3192, Japan; shuji@hama-med.ac.jp; 3Division of Biomedical Engineering, National Defense Medical College Research Institute, Saitama 359-8513, Japan; snaka@ndmc.ac.jp

**Keywords:** genome editing, CRISPR/Cas9, zygotes, postimplantation embryos, fetuses, primordial germ cells, spermatogonial stem cells, GONAD, TPGD-GEF, adeno-associated virus

## Abstract

The recently discovered clustered regularly interspaced short palindromic repeats (CRISPR)-associated protein 9 (Cas9) systems that occur in nature as microbial adaptive immune systems are considered an important tool in assessing the function of genes of interest in various biological systems. Thus, development of efficient and simple methods to produce genome-edited (GE) animals would accelerate research in this field. The CRISPR/Cas9 system was initially employed in early embryos, utilizing classical gene delivery methods such as microinjection or electroporation, which required ex vivo handling of zygotes before transfer to recipients. Recently, novel in vivo methods such as genome editing via oviductal nucleic acid delivery (GONAD), improved GONAD (*i*-GONAD), or transplacental gene delivery for acquiring genome-edited fetuses (TPGD-GEF), which facilitate easy embryo manipulation, have been established. Studies utilizing these techniques employed pregnant female mice for direct introduction of the genome-editing components into the oviduct or were dependent on delivery via tail-vein injection. In mice, embryogenesis occurs within the oviducts and the uterus, which often hampers the genetic manipulation of embryos, especially those at early postimplantation stages (days 6 to 8), owing to a thick surrounding layer of tissue called decidua. In this review, we have surveyed the recent achievements in the production of GE mice and have outlined the advantages and disadvantages of the process. We have also referred to the past achievements in gene delivery to early postimplantation stage embryos and germ cells such as primordial germ cells and spermatogonial stem cells, which will benefit relevant research.

## 1. Introduction

### 1.1. Genome-Editing Technology

Genome-editing techniques involve the use of sequence-specific nucleases, such as zinc-finger nucleases (ZFNs), transcription activator-like effector nucleases (TALENs), and clustered regularly interspaced short palindromic repeats (CRISPR)/CRISPR-associated protein 9 (CRISPR/Cas9) that make it possible to induce modifications in a predefined region of the genome [1]. These nucleases can induce double-strand breaks (DSBs) that are later repaired by the cellular machinery. In the absence of the donor (or template) DNA, the DSBs are repaired via nonhomologous end joining (NHEJ), which is error prone. NHEJ generates random insertions, deletions, or substitutions of nucleotides called indels at the break site. These indels often cause frameshift mutations, leading to the occasional formation of premature termination (stop) codons which cause protein expression failure through nonsense-mediated mRNA decay, a translation-dependent eukaryotic surveillance mechanism [2]. If a donor DNA containing longer genes (>1 kb) or single-stranded (ss) sequences (>200 bp) with homology to the target region is present, it will be introduced into the DSB site through homology-directed repair (HDR), a cellular mechanism that enables the precise recovery of the DSB. The most common form of HDR is homologous recombination (HR). This event is also called knock-in (KI) in which synthetic oligodeoxynucleotides (ODNs) 20–30 bp in size or sequences (>200 bp) are frequently used as DNA donors. Generally, KI is known to be more difficult to complete successfully than the induction of NHEJ-mediated indels. Furthermore, NHEJ occurs in nondividing as well as dividing cells but HDR occurs preferentially in dividing cells [3].

The CRISPR/Cas9 gene editing system requires two components: 1) a guide RNA (gRNA), comprised either of a duplex CRISPR RNA (crRNA)/trans-activating CRISPR RNA (tracrRNA) molecule or of single-guide RNA (sgRNA), a fusion between crRNA and tracrRNA, and 2) a Cas9 endonuclease [4,5,6]. The gRNA can bind to the specific DNA sequence together with Cas9. Once bound, the Cas9 nuclease causes double-stranded (ds) cleavage of the bound DNA at the portion 3 bp upstream of the protospacer adjacent motif (PAM, characterized by the sequence 5′-NGG-3′), which is recognized and bound by the Cas protein, which is subsequently repaired by various DNA repair mechanisms such as NHEJ and HDR. Thus, synthesis of gRNA is a prerequisite of the CRISPR/Cas9 system; the CRISPR/Cas9 system differs from the other genome-editing tools such as ZFNs and TALENs in this aspect. However, by virtue of being simple and cost-effective, the CRISPR/Cas9 technique is now widely used in various biological systems.

### 1.2. Developments in Genome-Edited (GE) Mice Production Technology

Efficient and safe delivery of components required for genome editing into germ-line-competent cells such as primordial germ cells (PGCs) and spermatogonial stem cells, preimplantation embryos such as zygotes (1-cell embryos) and 2-cell embryos and postimplantation embryos are a prerequisite for a successful genome-editing process.

Figure 1 is a schematic representation of the in vivo developmental process of murine embryos. Since embryogenesis occurs within oviduct and uterus, there are limited options for introducing genome-editing components into murine embryos. Genome-editing technology was initially experimented on zygotes isolated from oviducts of pregnant female mice or those obtained through in vitro fertilization (IVF) via preexisting methods, e.g., microinjection or in vitro electroporation (EP), as reviewed by Lino et al. [7] and Kaneko [8]. Viral vectors have also been used as alternatives to these classical methods. The most commonly used vectors are lentiviruses, adeno-associated viruses (AAVs), and adenoviruses. Lentiviruses and adenoviruses can transfect “denuded” preimplantation embryos from which zona pellucida (ZP) has been removed but not from those with intact ZP [9]. In contrast, AAVs can infect embryos with intact ZP, although the infection ability is dependent on the serotype [9,10]. As discussed later, two groups [9,10] have reported the successful production of GE mice through infection of preimplantation embryos with recombinant adeno-associated viruses (rAAVs) carrying genome-editing components. Recently, spermatogonial stem cells have been found to serve as a promising target for the production of GE mice [11,12,13,14,15]. This technology is called “spermatogonial stem cell-mediated transgenesis”, and the schematic representation is shown in Figure 2. Spermatogonial stem cells exist in the basal portion of seminiferous tubules and have an essential role in the production of sperm cells. These cells can be isolated and cultured in vitro and can surprisingly survive and produce sperm cells when transplanted inside a seminiferous tubule. The sperm cells derived from these transplanted spermatogonial stem cells can fertilize egg cells after intracytoplasmic sperm injection (ICSI). A few research groups have reported success in their attempts to produce GE mice by modifying the genome of spermatogonial stem cells [12,13,14,15]. The embryos resulting from in vitro gene delivery into isolated zygotes/2-cell embryos or ICSI using GE sperm are transferred to the reproductive tracts of pseudo-pregnant female mice to enable their development to full-term. This procedure, however, invariably entails ex vivo handling of cells and embryos.

In 2015, a novel in vivo approach called “genome-editing via oviductal nucleic acids delivery (GONAD)” was reported by Takahashi et al. [16]. This technique is performed by intraoviductal injection of a solution containing genome-editing reagents and subsequently by in vivo EP of the oviducts of a pregnant female at the 2-cell stage. The reagents injected into the lumen of the oviduct were therefore transferred into the 2-cell embryos floating within the oviduct under the influence of the electrical field created by EP. This approach yielded a successfully GE fetal offspring [16]. Yoon et al. [10] demonstrated that intraoviductal instillation of rAAVs into pregnant female mice resulted in the production of GE offspring. These results led to the inference that in vivo genome editing of preimplantation embryos is feasible. Recently, two more in vivo approaches for genome editing have been reported. One of the approaches involves in utero gene delivery into postimplantation fetuses, an approach used by a group that has demonstrated the efficacy of an intraamniotic injection (as shown in b on day 12.5 in Figure 1; the day when copulation plug is found is defined as day 0 of pregnancy) of rAAVs carrying genome-editing-related genes in the successful rescue of fetuses with lethal mutations [17]. The other approach involves tail-vein injection of a solution containing a plasmid (that confers expression of both Cas9 and gRNA) into a pregnant female at the mid-gestational stage. The reagents administered into the blood stream were transferred via the placenta to the fetal heart, resulting in successful genome editing of some fetal cardiac cells [18]. This technology was thereby named “transplacental gene delivery for acquiring GE fetuses (TPGD-GEF)”.

Detailed discussion of each genome-editing approach is discussed in the following sections.

## 2. Ex Vivo Delivery of Genome-Editing Components into Zygotes/2-Cell Embryos, PGCs, and Spermatogonial Stem Cells

### 2.1. Microinjection Technique

In 2013, knock-out (KO) mice were created by the CRISPR/Cas9 system in several laboratories [19,20,21,22,23,24,25,26,27,28]. Wang et al. [20] demonstrated that almost all the zygotes after microinjection with CRISPR/Cas9 components had mutated alleles and that at least three loci were simultaneously edited after a single shot of microinjection. However, almost all microinjection-based genome editing involved KO of a target gene via the NHEJ pathway. In addition, there have also been a number of reports where KI mice were successfully produced using the microinjection approach [21,29,30,31,32]. In these studies, a solution containing Cas9 mRNA, gRNA, and ssODNs (or plasmid DNA) as a donor template was used for pronuclear or cytoplasmic microinjection. KI of a small or long fragment into a target locus was achieved via HR at a CRISPR-directed DSBs. Notably, Gu et al. [33] have developed a novel CRISPR/Cas9-based method designated 2C-HR-CRISPR, by which up to 95% KI efficiency was achieved when mouse embryos were injected with CRISPR reagents containing fluorescent template DNA. Cytoplasmic microinjection of a solution containing an engineered Cas9 protein (making the donor fragment more accessible to the target sequence via biotin-streptavidin complexing) into 2-cell embryos having an extended G2-S phase caused a dramatic increase in HDR resulting in >10-fold increase in KI efficiency as compared to the preexisting methods.

Although microinjection of genome-editing reagents into zygotes is one of the most useful tools for generating genetically modified mice, it has advantages as well as disadvantages. The advantages of this technology include the delivery of known quantities of nucleic acids into a zygote irrespective of the type of zygote and the introduction of a large size cargo (carrying a gene of interest), which is a significant limiting factor when using viral vectors for gene delivery. The detailed protocols for microinjection-based genome editing have been described in Harms et al. [34] and Jacobi et al. [35].

### 2.2. EP Technique

EP is a method for delivering exogenous substances (e.g., DNA) into a cell by forming transient pores into the cell membranes under electrical stimulation in vitro and in vivo.

Since Kaneko et al. [36] first applied this technology to zygotes for producing GE animals, successful genome editing of mice [37,38,39,40,41,42] using this technology has become possible. Compared to the previously described microinjection-based technique, several zygotes (30 to 50) can be simultaneously genome edited using a square pulse generator (electroporator), a technique that does not require expensive micromanipulator systems. Detailed protocols for EP-based genome editing have been reported by Kaneko [43], Qin et al. [44], and Modzelewski et al. [45].

### 2.3. Gene Delivery to PGCs in the Genital Ridges

PGCs are derived from cells included in the epiblast, also called embryonic ectoderm (EEct), adjacent to the extra-embryonic ectoderm (ExEct) at day 6.5 of pregnancy. Such cells are then identified as cell clusters at the base of the allantois at day 7.25 of pregnancy [46,47,48]. They proliferate during the process of development and then move through the hindgut and dorsal mesentery towards the genital ridge at cay 10.5 of pregnancy [49,50]. By cay 13.5 of pregnancy, male PGCs exhibit mitotic arrest while female PGCs commence meiosis (reviewed by McLaren [51]). However, it remains unknown when and how functional germ cells are generated from these PGCs, which retain the ability to form pluripotent embryonic germ cells and teratocarcinomas under certain specific conditions [52,53,54].

Murine PGCs can be isolated from genital ridges at days 11.5–12.5 of pregnancy. Successful gene transfer into the isolated PGCs has been reported. For example, Watanabe et al. [55] evaluated in vitro EP, liposomal transfection, and calcium phosphate (CaPO_4_)-based gene delivery method to assess which tools can provide efficient transfection of mouse PGCs (isolated from gonads at day 11.5 of pregnancy). They found that EP causes serious damage to PGCs and that liposomal transfection resulted in poor transfection efficiency. In contrast, 18% of PGCs were successfully transfected with plasmid DNA co-precipitated with CaPO_4_. Unfortunately, these PGCs exhibited growth arrest in few days after being in culture. De Miguel et al. [56] first demonstrated that mouse PGCs can be successfully infected with lentiviruses. However, in this case, it is strictly required to develop a system allowing the genetically modified PGCs to transplant/survive within a juvenile genital ridge. Unfortunately, this is technically difficult because genital ridges are located on the posterior aspect of the embryo that is tightly surrounded by yolk sac. In this regard, Chuma et al. [57] provided an interesting approach for allowing isolated murine PGCs (from postimplantation embryos at day 8.5 of pregnancy) to survive in vivo. They transplanted genetically modified PGCs into the lumens of seminiferous tubules of immature and infertile male mice. The transplanted PGCs formed several foci with immature sperm within the seminiferous tubules. When the immature sperms were isolated from the foci and were subsequently subjected to ICSI, normal genetically modified offspring were successfully formed. Chuma et al. [57] concluded that PGC transplantation technique is useful for predicting the developmental potential of PGCs that are in vitro gene-engineered or recovered from embryos showing embryonic lethality. At present, there is no report on successful genome editing of isolated mouse PGCs.

Svingen et al. [58] developed a novel approach for genital ridge-targeted gene delivery in vitro. Complementary DNA (cDNA) expression constructs for the sex-determining region Y (*Sry*) protein, which is important for female-to-male sex-reversal, was injected into isolated genital ridge tissues (from day 11.5 fetuses), which were then subjected to magnetically induced transfection (magnetofection). This was followed by organ culture (culturing on agar blocks) of the transfected genital ridges to enable their further development under in vitro conditions. A fluorescent marker gene, concomitantly introduced into the constructs, at least 5 days after the organ culture was observed to be expressed. To our knowledge, there are no reports of successful genome-editing targeting PGCs in the genital ridges, as it is difficult to maintain the magnetofected genital ridges in vivo through transplantation into a fetal tissue.

Notably, Morohaku et al. [59,60] have recently developed a novel method for acquiring PGC-derived mature oocytes in vitro. They cultured mouse PGCs in fetal ovaries (at day 12.5 of pregnancy) under specific culture conditions that they have newly established, which allows follicle assembly and tight interaction between the oocytes and the follicular cells. It took about one month to allow PGCs to develop in vitro into mature oocytes. When these in vitro matured oocytes were subjected to IVF and the resulting fertilized eggs were transferred to recipient females, one hundred pups were successfully obtained from about 1000 matured oocytes. They concluded that this in vitro system could be useful for studying the mechanism of oogenesis at the molecular level and for preservation of female germ cells. This technology appears to be useful for creating GE animals when PGCs transfected with genome-editing components are allowed to develop in vitro into mature oocytes, which may lead to the production of GE offspring via fertilization with wild-type sperms.

### 2.4. Gene Delivery to Spermatogonial Stem Cells

Spermatogonial stem cells are a subpopulation of spermatogonia that are immature and capable of self-renewal but maintain the ability to differentiate into mature spermatozoa [61,62]. They can settle down and commence spermatogenesis after being transplanted into the seminiferous tubules of infertile recipient testes (male germ-cell transplantation) [63]. Mouse spermatogonial stem cells can be maintained in vitro at least for up to 2 years while maintaining their spermatogenic potential [64,65]. These cultured spermatogonial stem cells are later named as germline stem cells [64] and are shown to be useful for the production of genetically modified animals (including transgenic (Tg) and KO mice) [66,67,68,69], as shown schematically in Figure 2. The germline stem cells are first subjected to transfection with transgenes or targeting vectors. The resulting genetically modified germline stem cells are then subjected to male germ-cell transplantation, as Brinster and Zimmermann [63] first demonstrated. Subsequently, the in vivo developed genetically modified sperms are collected from the transplanted testes and subjected to ICSI to acquire offspring with the transgenes or targeted allele in their genome.

In 2015, genome-editing experiments with germline stem cells have been reported to be successful in mice [12,13,14,15]. Wu et al. [13] performed in vitro EP of CRISPR/Cas9 and exogenous wild-type 89-bp ssODNs into germline stem cells isolated from mutant mice (Crygc^−/−^)—carrying a single-nucleotide deletion in the crystalline gamma C (*Crygc*) gene—suffering from cataract. In the GE germline stem cells, the target sequence containing the deletion site was found to be successfully replaced by the ssODNs, as a result of NHEJ or HDR-mediated gene editing. ICSI of spermatogenic cells obtained after male germ-cell transplantation of these GE germline stem cells resulted in the production of cataract-free progeny. This experiment suggests that CRISPR-based gene correction is possible in germline stem cells. Sato et al. [12] performed genome-editing experiments (using double-nicking CRISPR/Cas9) in germline stem cells. The targeted locus was activated by stimulated by retinoic acid 8 (*Stra8*), which is known to be indispensable to spermatogenesis. However, the *Stra8*-targeted germline stem cells failed to differentiate into functional sperm cells after male germ-cell transplantation.

### 2.5. Infection of Preimplantation Embryos with AAVs

ZP is a layer of glycoproteins with several important functions and considered as a physical barrier for protection from any hazardous substance. As described previously, both lentiviruses and adenoviruses fail to transduce ZP-intact preimplantation embryos; however, they can transduce preimplantation embryos when those are placed in the perivitelline space (the space between the ZP and the membrane of the oocyte) or in a drop of medium containing ZP-free embryos [70,71]. Yoon et al. [10] first explored the possibility whether AAVs can transduce ZP-intact mouse zygotes and are useful as vehicles to edit an embryonic target gene, since the size of AAVs is smaller than lentiviruses and adenoviruses used for gene transfer experiments. Out of the 14 rAAV serotypes carrying enhanced green fluorescent protein (*EGFP*) transgene that were examined, all successfully transduced ZP-intact preimplantation embryos (morulae); serotype 6 (called rAAV-6) exhibited a high degree of fluorescence. Notably, infection with rAAV-6 was effective, irrespective of the mouse strain. Yoon et al. [10] next assessed the ability of rAAV-6 to induce CRISPR-based genome editing in a target gene in ZP-intact zygotes. They constructed rAAV6-*Cas9*, which carries *Cas9* gene under the control of mouse *U1a* small nuclear RNA promoter, and rAAV6-g*Tyr*, which contains an *EGFP* expression unit together with a gRNA expression unit targeted to the tyrosinase (*Tyr*) gene coding for the protein required for the synthesis of melanin. Incubation of zygotes in the presence of rAAV6-*Cas9* and rAAV6-g*Tyr* for 3 days (up to blastocysts) led to the production of blastocysts with 100% indels when treated with the highest concentrations of rAAVs (6 × 10^9^ genome copies). This finding suggests the importance of delivery of AAV particles inside the embryo where genome-editing components are released from AAV capsids, causing genome editing at the target locus.

Mizuno et al. [9] observed that, among the rAAVs that were tested, rAAV-6 showed the highest transduction efficiency when ZP-intact embryos were co-cultured with rAAVs; this was similar to what was shown by Yoon et al. [10]. Interestingly, both ZP-intact rat and bovine embryos were effectively infected by rAAVs. They also demonstrated successful CRISPR-based KI in mice. As shown in Figure 3, zygotes were first subjected to in vitro EP in the presence of a Cas9/gRNA complex called ribonucleoprotein (RNP) and then infected with rAAV-6 carrying a 1.8 kb green fluorescent protein (GFP) expression cassette flanked by two 100-bp *Rosa26* homology arms. The KI efficiency in the *Rosa26* locus was 15.5% and 6.3%, when assessed at the blastocyst and newborn stages, respectively.

Yoon et al. [10] examined whether rAAV-6-mediated infection of early embryos is possible in vivo. rAAV6-*Cas9* and rAAV6-g*Tyr* vectors were injected into the oviduct of pregnant female mice at day 0.5 of pregnancy. Out of the 29 pups that were obtained, 3 were found to have indels. All these mutated founder mice generated albino offspring indicating germline transmission. These results suggest that AAV particles can deliver CRISPR/Cas9 components to ZP-enclosed early embryos in vivo.

### 2.6. Gene Delivery to Postimplantation Embryos at Somite Stage

As shown in Figure 1, postimplantation embryos during days 5–9 of pregnancy are enclosed by decidua, a maternal uterine tissue which is important in blocking immunological attacks by maternal immune cells and in providing nutritional support to the embryo before establishment of the placenta. It is quite difficult to perform gene introduction through direct injection using a micropipette, since embryo visualization is hampered by the presence of decidua. Temporary in vitro cultivation of embryos at these stages is known to be feasible [72,73]. Therefore, many researchers performed gene delivery (via in vitro EP or injection of viral vectors) into dissected postimplantation embryos to assess gene function, cell movement, and cell lineage [74,75,76,77,78,79,80]. For example, Mellitzer et al. [76] microinjected a solution containing dsRNA (100 ng/μL) directed against orthodenticle homeobox 2 (*Otx2*) or forkhead box protein A2 (*Foxa2*) and fast green dye into the amniotic cavity of an isolated embryo (Day 7.5 of pregnancy), as shown in Figure 4A. Immediately after DNA injection, the embryo was subjected to in vitro EP. After 24–36 h in culture, the transfected embryos showed reduced expression of the target genes. Davidson et al. [77] transferred isolated day 7.5 egg cylinder stage embryos to a 10–15 μL drop of a solution containing plasmid, thus conferring targeted expression in the visceral endoderm (a germ layer surrounding the EEct and the ExEct of the gastrulation stage mouse embryo) and leaving it for 2–10 min at room temperature (left side in Figure 4B). The embryos were then subjected to in vitro EP (right side in Figure 4B). Pierreux et al. [78] performed gene delivery experiments using isolated day 8 embryos towards the endoderm (containing precursors of pancreas or liver) to trace the transgene expression. They first injected DNA into the prepancreatic or prehepatic territories, which had already been identified by co-injected cell markers of an embryo, and placed those in the EP cuvette prior to in vitro EP. These electroporated embryos were then subjected to whole embryo culture. They confirmed successful targeted transgene expression in the prepancreatic or prehepatic territories. Unfortunately, there are currently no reports on successful in vitro genome editing in the isolated egg cylinder- to somite-stage embryos.

## 3. In Vivo Delivery Targeted to Zygotes, 2-Cell Embryos, and Fetuses

### 3.1. GONAD/i-GONAD for Obtaining GE Animals

As previously mentioned, Takahashi et al. [16] performed GONAD in pregnant female mice at day 1.4 of pregnancy to induce mutations in the chromosomally integrated *EGFP* gene. This was done by direct injection of a solution (1–1.5 μL) containing *Cas9* mRNA and gRNA targeted towards *EGFP* cDNA into the lumen of an expanded area of the oviduct, called ampulla, under a dissecting microscope. After instillation, the entire oviduct was subjected to in vivo EP. Of all the mid-gestational fetuses obtained, 35% were fetuses losing EGFP fluorescence completely and 35% were those with reduced fluorescence. Sequencing analysis revealed that all the fetuses showing complete loss of fluorescence were bi-allelic KO fetuses and that fetuses showing reduced fluorescence were the ones that had mosaic mutations for the integrated *EGFP* gene. Gene delivery to the blastomere of an embryo is a possibility when GONAD is performed on day 1.4 of pregnancy (which corresponds to 2-cell stage), and this could be one of the causes for frequent generation of mosaic mutations. To circumvent these mosaic mutations, Ohtsuka et al. [81] modified the GONAD technique that was first developed by Takahashi et al. [16]. RNP was used instead of *Cas9* mRNA and gRNA, as the former is known to induce genome editing faster than the latter, and GONAD was performed on day 0.7 of pregnancy, a stage corresponding to late 1-cell embryos. By this stage, the cumulus cells—which surround the zygotes tightly at early 1-cell stage and are thought to hinder EP-mediated transfer of exogenous substances to an embryo—begin to detach. As a result, the efficiency of obtaining mice with indels was very high (~98%). Furthermore, this modified method created large deletions at a target gene as well as KI alleles. The resulting mutated traits were transmitted to the next generation. Based on these results, Ohtsuka et al. [81] renamed this modified technology, improved GONAD (*i*-GONAD). The *i*-GONAD was also proven to be effective in rats [82,83]. The detailed protocols for GONAD and *i*-GONAD have been reported by Gurumurthy et al. [84,85]. A review article regarding *i*-GONAD has been recently published by Ohtsuka and Sato [86]. In Figure 5, the procedure of *i*-GONAD is schematically shown together with the analysis of the embryos (morulae) isolated 2 days after *i*-GONAD (unpublished).

Infection of ZP-intact preimplantation embryos floating in the oviductal lumen has been proved to be possible by a simple intraoviductal instillation of rAAV-6-containing solution, similar to GONAD [10]. This approach has been referred to as “AAV-based GONAD”.

### 3.2. TPGD-GEF Technique

In 1995, Tsukamoto et al. [87] first demonstrated that a single shot intravascular delivery of plasmid (pSV40-CAT) complexed with the gene delivery reagent (Lipofectamine; 5-carboxyspermylglycine dioctadecylamide (DOGS)) into pregnant female mice ensures successful transfection of fetuses. This means that nucleic acids injected into the tail-veins of pregnant female mice can be delivered through the placental interface to the developing fetuses, where the expression of the exogenous gene occurs. Thus, this technique, which is also called TPGD, promises to provide a convenient means for researchers to study the effects of genes on embryonic development [88]. The possible mechanism of gene delivery to fetal tissues by TPGD is shown in Figure 6A.

Recently, Nakamura et al. [18] showed for the first time that the CRISPR/Cas9 system is capable of inducing indels in fetal cardiac cells. A solution containing an all-in-one type plasmid, pCGSap1-EGFP (capable of expressing both Cas9 and gRNA targeted to *EGFP* cDNA simultaneously; Figure 6B), complexed with FuGENE6 was intravenously injected into the tail-vein of pregnant wild-type female mice that had already been mated with male Tg mice carrying *EGFP* transgenes in a homozygous (Tg/Tg) state on day 12.5 of pregnancy (see Figure 6C). All the fetuses are then expected to be EGFP-expressing fetuses carrying the transgenes in a heterozygous (Tg/+) state. Thus, it is expected that TPGD-based delivery of the CRISPR system, targeted to *EGFP,* into these fetuses will cause reduction in the levels of EGFP fluorescence as a result of genome editing in the chromosomally integrated *EGFP* transgenes. Of the 24 fetuses isolated 2 days after TPGD, three exhibited reduced fluorescence in their heart (a vs. b in Figure 6D). A previous study has evaluated the preferential gene delivery into fetal hearts and circulatory systems [89]. Examination of the three isolated hearts for the presence of the transgene construct (*Cas9* gene) and possible indels revealed the presence of the *Cas9* gene and indels at the target loci of all the three samples. Notably, the indels were composed of normal and mutated cells (Figure 6D, panel (c)). These results suggest that this TPGD-based genome editing is sufficient to cause a defect in the embryonic heart due to mosaic mutations and has the potential to be used for the production of cardiovascular disease models and for fetal gene therapy in the background of congenital heart diseases. This study also emphasizes the potential of the pCGSap1-based plasmid DNA to act as a nonviral gene delivery cargo for inducing in vivo genome modifications at a target locus.

### 3.3. In Utero Gene Delivery

Most studies on gene delivery to egg cylinder (day 7.5) or somite (day 8.5) stage embryos have focused on ex vivo manipulation of the isolated embryos. However, a study by Sheehy et al. [90] is an exception. In this study, 3 μL of a solution containing miR-452 antagomir (specific inhibitor to block the function of microRNA-452) was injected using a 35-gauge needle into the space between the embryo and the yolk sac through the decidua of a day 8.5 embryo (see day 8.5 in Figure 1), under anesthesia. After surgery, the embryos were allowed to develop to mid-gestational stages and were then sampled for analysis. It was demonstrated that the neural crest cell-specific reduction of microRNA resulted in the generation of abnormal fetuses, which were lacking in craniofacial cartilaginous structures. However, currently, there is no report on in vivo successful genome editing in the somite-stage embryos.

Experiments using in utero gene delivery methods are usually performed on mid- to late-gestational fetuses (from days 9–18 of pregnancy), probably because they are relatively easier to be identified by visual observation (see day 12.5 in Figure 1). In these experiments, viral vectors, such as AAVs, lentiviruses, and adenoviruses have been involved in infecting the fetuses at days 9–15 of pregnancy. Experiments using in utero gene delivery of genome-editing components have been performed by several research groups [17,91,92,93,94]. Ricciardi et al. [93] demonstrated that intra-amniotic administration of polymeric, biodegradable nanoparticles derived from poly (lactic-co-glycolic acid) containing triplex-forming peptide nucleic acids and donor ssDNAs into mouse fetuses carrying mutations in the β-globin gene, results in the offspring being rescued from a disease similar to human β-thalassemia. These rescued offspring showed sustained elevation of blood hemoglobin levels, reduced number of reticulocytes, disappearance of splenomegaly, and longer survival. Alapati et al. [17] have recently used CRISPR technology to correct a lung-disease-causing gene (called *SftpcI73T*)—a mutated version of the pulmonary-associated protein C gene—that codes for a protein that helps prevent the lung from collapsing when emptied. Embryonic expression of *SftpcI73T* is known to cause severely diffused parenchymal lung damage leading to the early death of affected individuals. Alapati et al. [17] performed intra-amniotic administration of CRISPR reagents into day 16 fetuses carrying the *SftpcI73T* gene. Gene editing was successful in the target cells of the lungs of 20% of the mice born. In addition, the report states that the genetic alteration was not observed in the germ cells, indicating that genetic changes cannot be transmitted to the offspring. This study has the potential to aid the development of novel fetal gene-editing methods that target human monogenic disorders.

## 4. Conclusions

In this review, we have described the applicability of CRISPR/Cas9 system for producing GE animals via ex vivo manipulation of embryos and germ cells as well as direct in vivo gene delivery to pregnant female mice. While the CRISPR/Cas9 technique is considered to be one of the promising tools for inducing genome modification at the desired target loci, the possibility of off-target mutations is a limitation of the technique. This results in concerns regarding the safety of this technique when used for gene therapeutic studies in humans. The use of newly engineered nuclease variant called “SaCas9-HF” which is originally derived from Cas9 nuclease of the *Staphylococcus aureus* bacteria (SaCas9) promises to overcome this limitation. Wild-type SaCas9 has been shown to reduce the off-target activities by about 90% for gene sequences that are highly similar and are therefore susceptible to mutations [95]. This modification to the CRISPR/Cas9 system will thus prove beneficial for producing GE animals with reduced off-target effects.

Though the CRISPR/Cas9-based genome-editing technique is highly advanced, various other technologies for producing GE experimental animals have also been developed. These include zygote microinjection, in vitro EP of zygotes, GONAD, *i*-GONAD, TPGD-GEF, germ cell-based genome editing using germline stem cells, and in utero gene delivery. As summarized in Table 1, each of these technologies has specific characteristics. However, it has been long thought to be difficult to manipulate embryos (at days 6–8 of pregnancy) in vivo because they are surrounded by decidua. Fortunately, Sheehy et al. [90] have suggested that in vivo gene delivery to embryos at these stages is indeed possible via glass capillary-based injection of a nucleic acid-containing solution under a dissecting microscope. Furthermore, in vitro gene delivery to isolated embryos (days 6–7) is now possible. If a more sophisticated technology for in vivo gene delivery to these postimplantation embryos is developed in the near future, it may be possible to perform genome editing of these embryos in vivo. This point is also true for isolated PGCs which, when cultured in vitro, can develop into mature sperm cells or oocytes under appropriate environmental conditions. This means that it is possible to create GE animals that have been derived from PGCs engineered in vitro.

## Figures and Tables

**Figure 1 cells-09-00799-f001:**
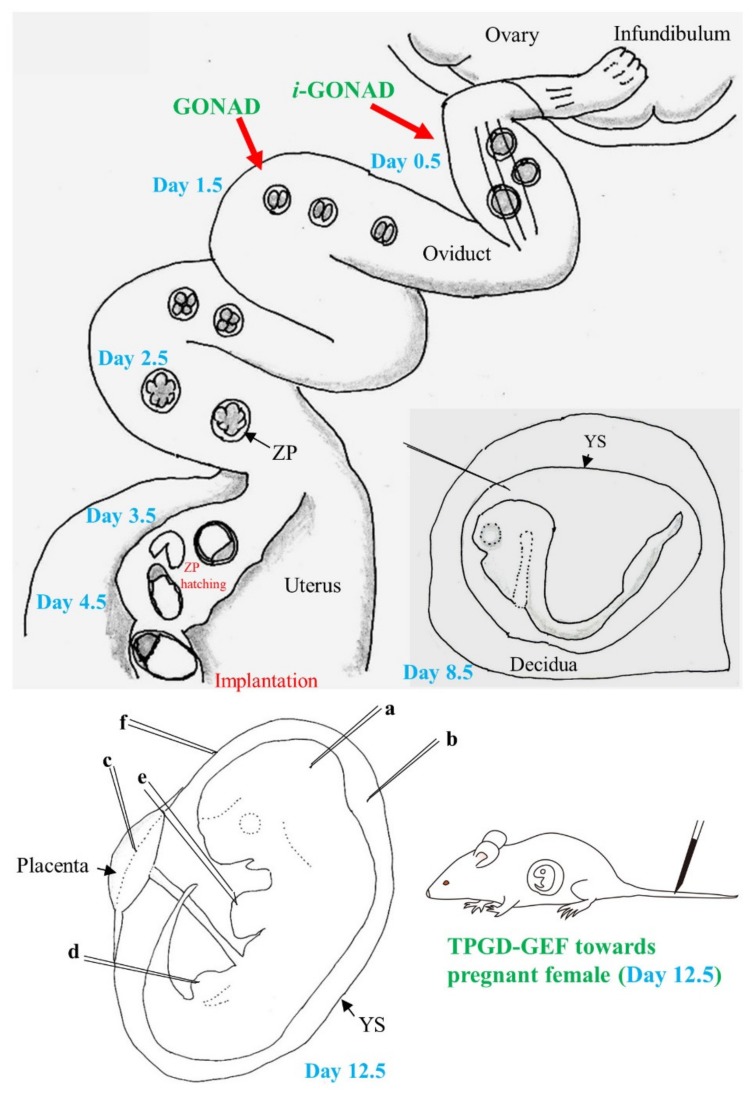
Schematic representation of preimplantation (days 0.5 to 4.5; day 0 of pregnancy is defined as the day a vaginal plug is found) and postimplantation (days 5.5 to 9.5 and 11.5) development of mice: During the preimplantation stage, zygote (1-cell embryo) (at day 0.5), 2-cell embryo (at day 1.5), 8- to 16-cell embryo (at day 2.5), early blastocyst (at day 3.5), and late blastocyst (at day 4.5) float in the oviductal lumen or uterine horn. Embryos at days 0.5 to 4.5 have zona pellucida (ZP), but embryos at day 4.5 begin to escape from ZP, which is called “ZP hatching”, and form early egg cylinder containing epiblast after implantation. In vivo genome editing is possible after intraoviductal instillation of a solution containing the genome-editing components and subsequent in vivo electroporation (EP), known as the “genome-editing via oviductal nucleic acids delivery (GONAD)” (for 2-cell embryos) and “improved GONAD (*i*-GONAD)” (for zygotes) techniques. On day 8.5, embryos commence somite formation with the formation of blood-island and a beating heart. Furthermore, closure of neural tube commences. Notably, embryos at days 5 to 8 are surrounded by decidua, which does not allow visualization of embryos upon surgical dissection of the uterus. Transfection of day 8.5 embryos is possible when a nucleic acid-containing solution is injected into the internal portion of decidua through the yolk sac (YS) using a micropipette. On day 12.5, the fetus (embryo) is visible through the YS upon surgical dissection of the uterus under a dissecting microscope. Thus, it is possible to administer intrabrain (**a**), intraamniotic (**b**), intraplacental (**c**), intramuscular (**d**), intracardiac (**e**) and intravitelline (**f**) injections using a micropipette for in utero gene delivery. Tail-vein injection of a solution containing genome-editing components into pregnant female mice is also a useful in vivo approach to induce genome editing in day 12.5 fetuses, which is called “transplacental gene delivery for acquiring genome-edited fetuses (TPGD-GEF)”.

**Figure 2 cells-09-00799-f002:**
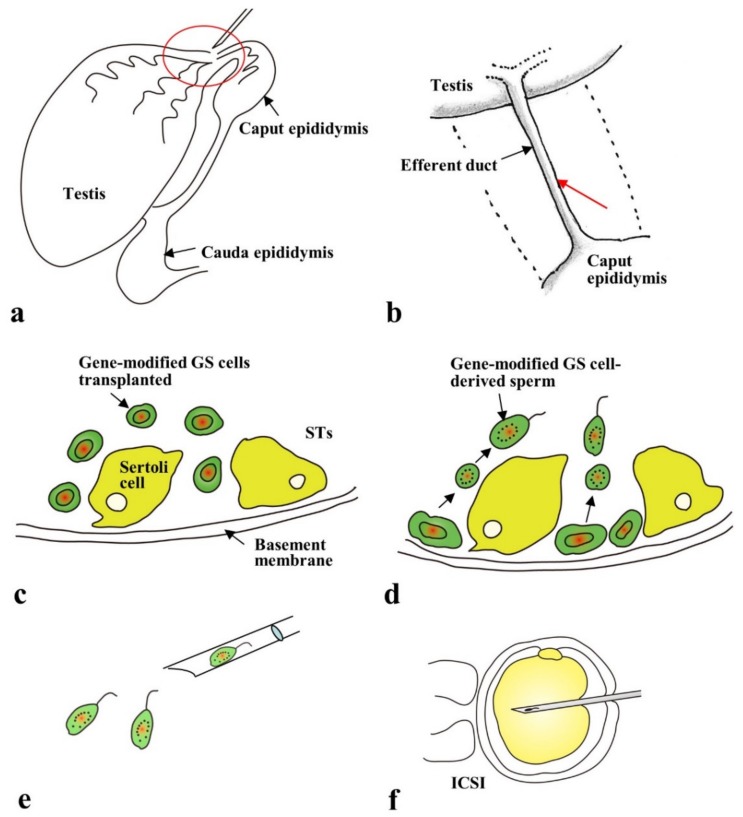
Transplantation of gene-engineered germline stem (GS) cells into the lumen of seminiferous tubules (STs) of a sterile testis and subsequent intracytoplasmic sperm injection (ICSI) of the GS cell-derived sperm for production of genetically modified mice, schematically shown: (**a**) Transplantation of the gene-engineered GS cells via rete testis inside the STs of a testis that has been pretreated with busulfan to eliminate endogenous spermatogonial stem cells. The portion surrounded by a red circle is shown in **b** for more detailed explanation. (**b**) Transplantation of cells by insertion of a glass micropipette into an efferent duct to an orientation shown by the red arrow. (**c**) Gene-engineered GS cells just after introduction into the lumen of the STs. (**d**) Differentiation of gene-engineered GS cells. The transplanted gene-engineered GS cells move from the luminal compartment to the basement membrane where spermatogonia are located. Then, they survive and proliferate, while also producing immature sperm. (**e**) Gene-engineered GS cell-derived immature sperm isolated from the transplanted testis, which will be used for ICSI. (**f**) ICSI using GS-derived immature sperm. The ICSI-treated oocytes are then allowed to develop further through embryo transfer to the oviducts of pseudo-pregnant females to generate the gene-engineered pups.

**Figure 3 cells-09-00799-f003:**
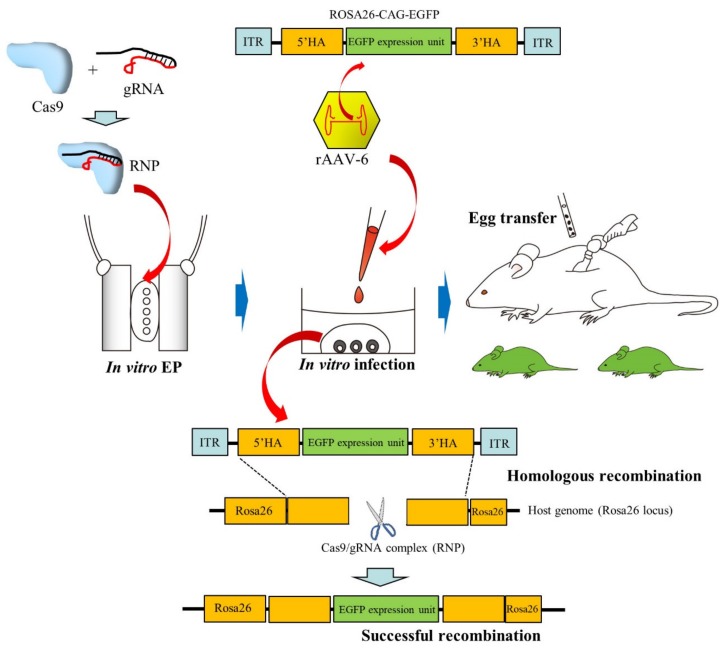
Schematic representation of recombinant adeno-associated virus-6 (rAAV-6)-based production of knock-in (KI) mice, according to the method of Mizuno et al. [9]: Zygotes are first subjected to in vitro electroporation (EP) in the presence of the Cas9/gRNA complex (called ribonucleoprotein, RNP) and then subjected to infection with rAAV-6 carrying donor sequence (ROSA26-CAG-EGFP). After this, it is expected that CRISPR/Cas9-mediated KI of the donor sequence into the *ROSA26* locus occurs in the infected zygotes. These treated zygotes are subsequently transferred to the oviducts of pseudopregnant females to generate KI pups.

**Figure 4 cells-09-00799-f004:**
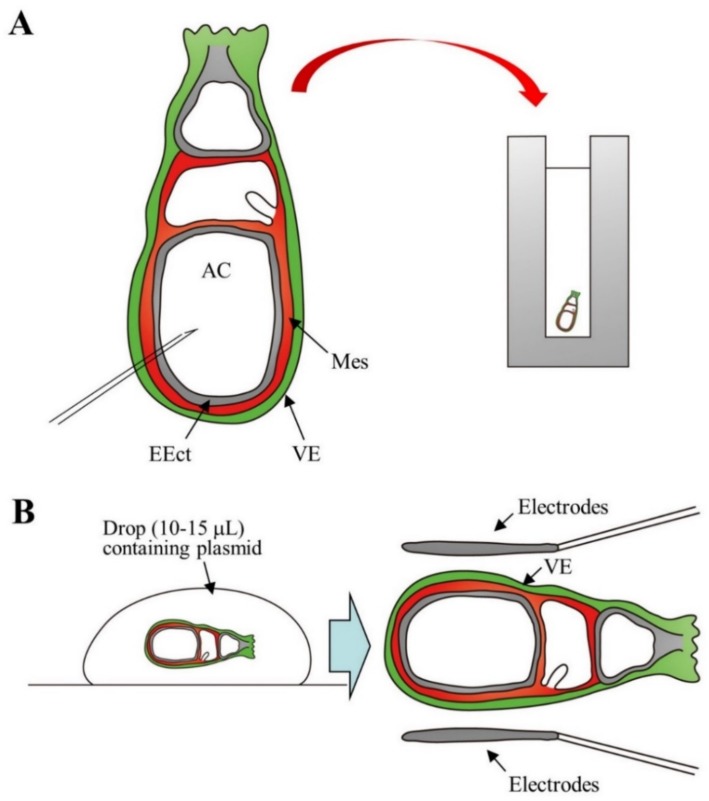
Schematic representation of gene delivery towards the isolated egg cylinders-stage embryos (day 7.5; day 0 of pregnancy is defined as the day a vaginal plug is found): (**A**) Injection of nucleic acid into the amniotic cavity (AC) and subsequent in vitro electroporation (EP). Prior to pulse application, an embryo is placed in a cuvette. Abbreviations: EEct, embryonic ectoderm; Mes, mesoderm; VE, visceral endoderm. (**B**) Transfection of VE cells by simple incubation in a drop containing plasmid DNA and subsequent in vitro EP: Prior to pulse application, the electrodes are positioned parallel to the embryo.

**Figure 5 cells-09-00799-f005:**
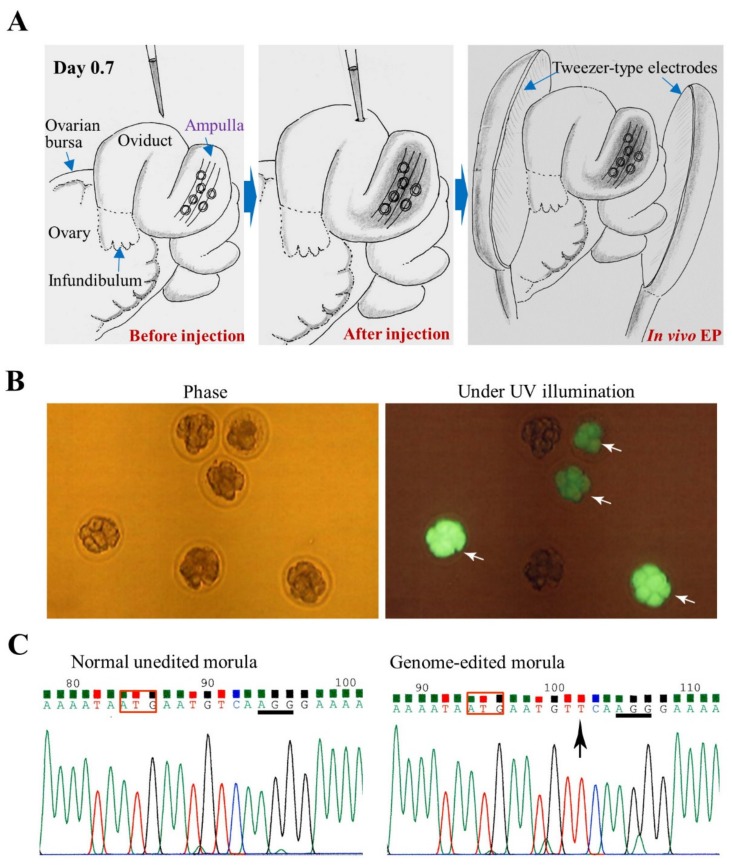
*i*-GONAD in mice for targeting *GGTA1* locus coding for α-1,3-galactosyltransferase, an enzyme capable of synthesizing cell-surface α-Gal epitope: (**A**) Schematic illustration of the *i*-GONAD procedure. Intraoviductal instillation of a solution containing RNP (Cas9/gRNA complex) and fluorescein isothiocyanate (FITC)-labelled dextran 3 kDa (a fluorescent marker for monitoring successful *i*-GONAD) through the oviductal wall is first carried out on day 0.7 of pregnancy (day 0 of pregnancy is defined as the day a vaginal plug is found). Then, the entire oviduct is subjected to in vivo electroporation (EP) using tweezer-type electrodes. (**B**) Isolated morulae 2 days after *i*-GONAD: There are some embryos showing FITC fluorescence (indicated by arrows). (**C**) Direct sequencing of PCR products derived from an *i*-GONAD-treated embryo shown in **B**. The sequences corresponding to the region around ATG (boxed) in *GGTA1* in normal unedited and successfully edited embryos are shown. Arrow indicate insertion of nucleotide in front of the protospacer adjacent motif (PAM; underlined).

**Figure 6 cells-09-00799-f006:**
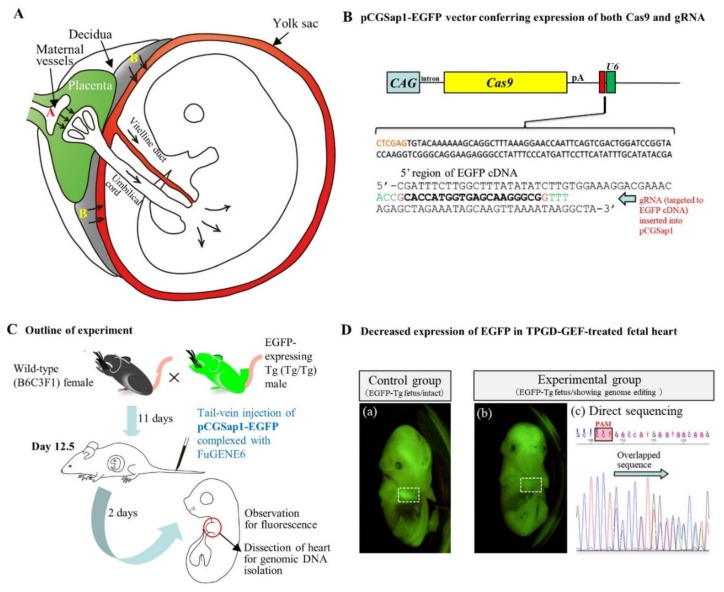
Transplacental gene delivery (TPGD) on day 12.5 (day 0 of pregnancy is defined as the day a vaginal plug is found): (**A**) Hypothetical mechanism of TPGD suggested by Kikuchi et al. [89] and Nakamura et al. [88]. When the TPGD is performed at day 12.5 (the time when placental circulation is established), the intravenously injected plasmid DNA/lipid complexes may be transferred via at least two routes from maternal blood to the fetus. One route is via placenta to the embryo, as indicated by the arrows (area A). Injected plasmid DNA is transferred beyond the blood-placenta barrier and enters the umbilical cord. The other route is from the decidua to the yolk sac, as depicted by the arrows (area B). Some DNA become trapped in the yolk sac and is transferred to the embryo via the vitelline circulation. (**B**) Structure of pCGSap1-EGFP, an all-in-one type plasmid, capable of expressing both Cas9 and gRNA (targeted to *EGFP* cDNA) simultaneously: This plasmid has been used for genome editing of EGFP-expressing fetal cardiac cells in TPGD-GEF (cited from Nakamura et al. [88] under permission of MDPI, publisher of the *International Journal of Molecular Science*). (**C**) Schematic representation of the experimental outline of TPGD-GEF. At day 12.5 of pregnancy, a solution containing plasmid DNA complexed with gene delivery reagent (i.e., FuGENE6) was intravenously administered to the pregnant female mice. Two days after in vivo transfection, fetuses were dissected to check the presence of the introduced DNA and its expression. (**D**) Decreased expression of EGFP-derived fluorescence in the heart of a TPGD-GEF-treated fetus. (**a**) Intact control fetus. The heart (enclosed by white dashed box) exhibited strong fluorescence. (**b**) TPGD-GEF-treated fetus exhibiting reduced fluorescence in its heart (enclosed by white dashed box). (**c**) Sequence analysis of PCR products (corresponding to the 5′ region of the *EGFP* sequence) from the fetus (**b**). Overlapping electrophoretograms (indicated by arrows) immediately upstream of the protospacer adjacent motif (PAM) indicate mosaic pattern of mutations.

**Table 1 cells-09-00799-t001:** Summary of the characteristic properties of the clustered regularly interspaced short palindromic repeats (CRISPR)/associated protein 9 (Cas9)-based genome-editing techniques used for embryos, fetuses, and germ cells in mice.

Delivery mode(s)	Method(s)	Target Embryos, Fetuses and Cells	Equipment	Notes	References
Ex vivo	Microinjection	Zygotes 2-Cell embryos	Requires micromanipulator system	Requires egg transfer to allow further development of the treated embryos	[19,20,21,22,23,24,25,26,27,28,29,30,31,32,33,34,35]
Ex vivo	In vitro EP	Zygotes Germ stem cells	Requires electroporator and micromanipulator system	Requires ICSI and egg transfer to allow further development of the treated embryos	[12,13,14,15,36,37,38,39,40,41,42,43,44,45]
Ex vivo	Transduction by viral vectors	Zygotes	No special equipment	Requires egg transfer to allow further development of the treated embryos	[9,10]
In vivo	GONAD/*i*-GONAD/AAV -based GONAD	Zygotes 2-cell embryos	Requires electroporator but no special equipment in the case of AAV-based GONAD	No recipients are required	[10,16,81,84,85,86]
In vivo	TPGD/TPGD-GEF	Mid-gestational fetuses	No special equipment	No recipients are required	[18,88]
In vivo	In utero gene delivery using viral vectors or in vivo EP	Mid-gestational fetuses	Requires electroporator in the case of using nonviral vectors	No recipients are required	[91,92,93,94]

Abbreviations: AAV, adeno-associated virus; EP, electroporation; GONAD, Genome-editing via oviductal nucleic acids delivery; *i*-GONAD, improved GONAD; ICSI, intracytoplasmic injection of sperm; TPGD, transplacental gene delivery; TPGD-GEF, transplacental gene delivery for acquiring genome-edited fetuses.

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
