# Peer review of "Recent Advances and Future Perspectives of In Vivo Targeted Delivery of Genome-Editing Reagents to Germ cells, Embryos, and Fetuses in Mice"

_cells, 2020, doi:10.3390/cells9040799_

Round 1
Reviewer 1 Report
Dear Cells Editorial Office,
With respect to Manuscript ID: cells-745153
Title: Recent Advances and Future Perspectives of In Vivo Targeted Delivery
of Genome Editing Reagents to Germ cells, Embryos, and Fetuses in Mice
Authors: Masahiro Sato, Shuji Takabayashi, Eri Akasaka, Shingo Nakamura
Please note the following Specific Comments
Page 2
Lines 46-48
The authors state
“These indels often cause frameshift mutations, leading to abnormal protein formation or absence of protein formation in case of a premature stop codon, which would prematurely terminate the protein synthesis.”
This information is incorrect. Please correct this statement and provide a citation. Nonsense mediated mRNA decay is the mechanism by which premature termination (stop) codons cause protein expression failure. Premature termination codons do not prematurely terminate protein synthesis when they are located 50-55 nt upstream of an exon-exon boundary.
Page 2
Lines 57-58
The authors state
“The CRISPR/Cas9 gene editing system requires two components, a short (20 bp) RNA sequence called guide RNA (gRNA) and Cas9 endonuclease [3−6].”
This information is incorrect. gRNA is made up of a duplex RNA molecule containing crRNA and tracRNA. Alternatively a single guide RNA (sgRNA) is made with a linker between the tracRNA scaffold and the target specific crRNA. Please read the citations you made and rewrite these lines to accurately describe both duplex gRNA and sgRNA. If necessary, substitute a scientific citation that describes gRNA and sgRNA such as Doudna and Charpentier (2014).
Genome editing. The new frontier of genome engineering with CRISPR-Cas9.
Doudna JA, Charpentier E.
Science. 2014 Nov 28;346(6213):1258096. doi: 10.1126/science.1258096. Review.
PMID: 25430774
Page 2
Lines 59-61
The authors state
“Once bound, Cas9 cleaves one of the double-stranded (ds) chromosomal DNA at the portion 3 bp upstream of the protospacer adjacent motif (PAM), and subsequently induces a DSB at the cleaved site, which is then repaired by NHEJ.”
Please rewrite this sentence so it is accurate. Please explain the requirement for the PAM for Cas9 cleavage. Please explain that Cas9 causes a double stranded DNA break, not a single stranded break. Cas9 only breaks DNA, it does not repair DNA by NHEJ. Please refer to the review above for an explanation of the mechanism by which the sgRNA-Cas9 ribonucleoprotein complex causes a chromosome break. Please explain the mechanism in your own words clearly and accurately.
Page 7
Line 214
The authors state
“How is development of immature oocytes derived from female PGCs?”
This sentence has no meaning in the English language. Please clarify and re-state.
Page 7
Caption for Figure 2
The caption for Figure 2 is hopelessly garbled.
Please replace the caption for Figure 2 with a new caption that completely explains Figure 2 and all panels in Figure 2.
Page 12
Line 339
Please replace the word “loosing” with the word “losing”. Although these two words are spelled in a similar fashion they have completely different meanings.
Author Response
Comments and Suggestions for Authors
Please note the following
Specific Comments
1) Page 2 Lines 46-48
The authors state “These indels often cause frameshift mutations, leading to abnormal protein formation or absence of protein formation in case of a premature stop codon, which would prematurely terminate the protein synthesis.” This information is incorrect. Please correct this statement and provide a citation. Nonsense mediated mRNA decay is the mechanism by which premature termination (stop) codons cause protein expression failure. Premature termination codons do not prematurely terminate protein synthesis when they are located 50-55 nt upstream of an exon-exon boundary.
Answer: Thank you for your helpful suggestion. As suggested, we changed this portion, as follows: “These indels often cause frameshift mutations, leading to the occasional formation of premature termination (stop) codons which cause protein expression failure through nonsense-mediated mRNA decay, a translation-dependent eukaryotic surveillance mechanism [2]”. Please see lines 46–49 in the revised text.
2) Page 2 Lines 57-58
The authors state “The CRISPR/Cas9 gene editing system requires two components, a short (20 bp) RNA sequence called guide RNA (gRNA) and Cas9 endonuclease [3−6].” This information is incorrect. gRNA is made up of a duplex RNA molecule containing crRNA and tracRNA. Alternatively a single guide RNA (sgRNA) is made with a linker between the tracRNA scaffold and the target specific crRNA. Please read the citations you made and rewrite these lines to accurately describe both duplex gRNA and sgRNA. If necessary, substitute a scientific citation that describes gRNA and sgRNA such as Doudna and Charpentier (2014). Genome editing. The new frontier of genome engineering with CRISPR-Cas9. Doudna JA, Charpentier E. Science. 2014 Nov 28;346(6213):1258096. doi: 10.1126/science.1258096. Review. PMID: 25430774
Answer: As suggested, we have rewritten this portion as follows: “The CRISPR/Cas9 gene editing system requires two components: 1) a guide RNA (gRNA), comprised either of a duplex CRISPR RNA (crRNA)/trans-activating CRISPR RNA (tracrRNA) molecule, or of single-guide RNA (sgRNA), a fusion between crRNA and tracrRNA, and 2) a Cas9 endonuclease [4−6].” Please see lines 57–60 in the revised text.
3) Page 2 Lines 59-61
The authors state “Once bound, Cas9 cleaves one of the double-stranded (ds) chromosomal DNA at the portion 3 bp upstream of the protospacer adjacent motif (PAM), and subsequently induces a DSB at the cleaved site, which is then repaired by NHEJ.” Please rewrite this sentence so it is accurate. Please explain the requirement for the PAM for Cas9 cleavage. Please explain that Cas9 causes a double stranded DNA break, not a single stranded break. Cas9 only breaks DNA, it does not repair DNA by NHEJ. Please refer to the review above for an explanation of the mechanism by which the sgRNA-Cas9 ribonucleoprotein complex causes a chromosome break. Please explain the mechanism in your own words clearly and accurately.
Answer: As suggested, we have rewritten this portion as follows: “Once bound, the Cas9 nuclease causes double-stranded (ds) cleavage of the bound DNA at the portion 3 bp upstream of the protospacer adjacent motif (PAM, characterized by the sequence 5’-NGG-3’), which is recognized and bound by the Cas protein, which is subsequently repaired by various DNA repair mechanisms such as NHEJ and HDR.” Please see lines 60–64 in the revised text.
4) Page 7 Line 214
The authors state “How is development of immature oocytes derived from female PGCs?” This sentence has no meaning in the English language. Please clarify and re-state.
Answer: As suggested, this sentence appears to be unnecessary. We have rewritten this portion as follows: “Notably, Morohaku et al. [59,60] have recently developed a novel method for acquiring primordial germ cell-derived mature oocytes in vitro.” Please see lines 225–226 in the revised text.
Page 7 Caption for Figure 2
The caption for Figure 2 is hopelessly garbled. Please replace the caption for Figure 2 with a new caption that completely explains Figure 2 and all panels in Figure 2.
Answer: As suggested, we have redrawn Figure 2, and rewritten its figure legend. Please see the updated Figure 2 and its legend in the revised text.
Page 12 Line 339
Please replace the word “loosing” with the word “losing”. Although these two words are spelled in a similar fashion they have completely different meanings.
Answer: As suggested, we have replaced the word “loosing” with the word “losing” (please see line 363 in the revised text).

Reviewer 2 Report
The review requires some editing improve readability.
- First, being the topic inherently complex, it would be better to reduce the acronyms. Each time an acronym is mentioned, the reader must go back to the extensive list, and this results in a loss of time and in a loss of focus on the scientific issue presented. For instance AAVs, AC, AVs, GOI, GR, GS, ICSI, LVs, MI, ODNs, PGCs, PNAs, rAVV-6,SFTPC, SSCs, STs, Stra-8, TPGD, TPCD-GEF, VF, YS, ZFNs, ZP could be spelled out when occurring in the text, and left as abbreviations only in the Figures (with the list in the legend)
- Introduction, section 1.2, Developments in Genome-Edited Mice....: here (line 68) the term "fetuses" should be corrected to "post-implantation embryos". In this regard, please check accurately the Ms, including Figures, to correct fetus, fetal, or fetuses to embryo, embryonic or embryos wherever necessary
- section 2.1 MI technique: line 143, it seems more appropriate "Gu et al developed a novel CRISPR/Cas9-based method designated....." Line 152: "and the introduction"; Line 155: the phrase is cumbersome, please rewrite. Line 171-172: "called an electroporator....." may just be "(electroporator), a technique that does not require expensive micromanipulator systems." (delete "thereby improving...."). Lines 172-173: same detailed protocol or detailed protocols? Correct accordingly.
- 2.3 Gene delivery to PGCs.... The first 3 lines (176-78): better to correct to " PGCs are derived from cells included in the epiblast, also called embryonic ectoderm, .......Such cells..... Line 187: "method to assess" Line 189: "damage" not "damages". Line 195: "GRs are located on the posterior aspect of the embryo". Line 195: "Svingen et al developed... ". line 207: "injected into isolated..." Line 214: rephrase the question.
- 2.4 Gene Delivery to SSCs. Line 232: "that are immature and capable of self-renewal, but maintain the ability to differentiate..."Lines 234-235: "which is called" not necessary. Just put (male germ-cell transplantation"). Line 253: "stimulated by stimulated by retinoid acid 8..": rephrase.
- 2.5. Infection of preimplantation .... Lines 265-266: The zona pellucida is a layer of glycoproteins with several important functions. the statement that is a glycoprotein protecting from hazardous substances is not correct. Line 268: " "(the space between the zona pellucida and the membrane of the oocyte)". "in a medium...": "in a drop of medium containing...?". line 279: "to the the tyrosinase (Tyr) gene coding". Line 285: rephrase the sentence.
- 2.6. Gene delivery to post-implantation....Line 308: "in blocking immunological attacks by maternal immune cells"; Line 309: "support to the embryo"; line 311: "embryo visualization"; "Temporal: correct to Temporary; Line 330: "on successful in vitro genome..."
- 4. Concluding remarks: The last paragraph (lines 483-493) should be brought up at the beginning, integrating the concepts with those expressed in lines 458-459. The clinical applications here lack of a reference and are in contrast with the concepts expressed below. Line 463: "each of these technologies has specific characteristics." Line 467-468: "than the above mentioned techniques " is not necessary;
- Table 1: Title: Summary of the characteristic properties of the ....editing techniques... Column headings: "Delivery mode(s)"; "Method(s)", "Equipment and others???" What do you mean with "others"?; . Please reduce the text in the columns to the essential.
- Figure 2: Please rewrite the legend. What does "Table 2" refer to?
- Figure 3, legend: "Schematic representation of..." line 263: "to generate KI pups".
- Figure 4: Provide a better annotation for day 7.5 embryonic stage. " Drop containing plasmid DNA" Legend: schematic representation of....
- Figure 5, legend, line 367: "through the oviductal..."
- Figure 6, Legend: on day = at day; line 390. "is transferred to the embryo via the vitelline..." Line 393: "MDPI, publisher of...."
Author Response
Comments and Suggestions for Authors
The review requires some editing improve readability.
- First, being the topic inherently complex, it would be better to reduce the acronyms. Each time an acronym is mentioned, the reader must go back to the extensive list, and this results in a loss of time and in a loss of focus on the scientific issue presented. For instance AAVs, AC, AVs, GOI, GR, GS, ICSI, LVs, MI, ODNs, PGCs, PNAs, rAVV-6, SFTPC, SSCs, STs, Stra-8, TPGD, TPCD-GEF, VF, YS, ZFNs, ZP could be spelled out when occurring in the text, and left as abbreviations only in the Figures (with the list in the legend).
Answer: Thank you for the helpful suggestion. As suggested, we have ceased using the abbreviations you’ve listed in the text where possible, except for AAVs, rAAV-6, ICSI, ODNs TPGD, TPCD-GEF, PGC and ZP, which are abbreviations of important terms frequently used in the text, as well as generally employed in the field of reproductive biology and molecular biology.
- Introduction, section 1.2, Developments in Genome-Edited Mice....: here (line 68) the term "fetuses" should be corrected to "post-implantation embryos". In this regard, please check accurately the Ms, including Figures, to correct fetus, fetal, or fetuses to embryo, embryonic or embryos wherever necessary
Answer: As suggested, we have replaced the term "fetuses" with the term "post-implantation embryos". Please see lines 73, 112, 114, and 331 in the revised text.
- section 2.1 MI technique: line 143, it seems more appropriate "Gu et al developed a novel CRISPR/Cas9-based method designated....." Line 152: "and the introduction"; Line 155: the phrase is cumbersome, please rewrite. Line 171-172: "called an electroporator....." may just be "(electroporator), a technique that does not require expensive micromanipulator systems." (delete "thereby improving...."). Lines 172-173: same detailed protocol or detailed protocols? Correct accordingly.
Answer: As suggested, we have made corrections to the following:
For line 143: “--- have developed a novel CRISPR/Cas9-based method designated 2C-HR-CRISPR, by which up to 95% KI efficiency was achieved when mouse embryos were injected with CRISPR reagents containing fluorescent template DNA.” Please see lines 152–155 in the revised text.
For line 152, containing the phrase “and the introduction”, please see line 162 in the revised text.
For line 155: This portion has been deleted. Please see line 163 in the revised text.
For lines 171–172, we have revised the sentence to: “... (electroporator), a technique that does not require expensive micromanipulator systems." Please see lines 180–181 in the revised text.
For lines 172–173: “Detailed protocols for EP-based genome editing have been ...”, please see lines 181–182 in the revised text.
- 2.3 Gene delivery to PGCs.... The first 3 lines (176-78): better to correct to "PGCs are derived from cells included in the epiblast, also called embryonic ectoderm, .......Such cells..... Line 187: "method to assess" Line 189: "damage" not "damages". Line 195: "GRs are located on the posterior aspect of the embryo". Line 195: "Svingen et al developed... ". line 207: "injected into isolated..." Line 214: rephrase the question.
Answer: As suggested, we have made the following corrections:
For lines 176–178, “PGCs are derived from cells included in the epiblast, also called embryonic ectoderm (EEct), adjacent to the extra-embryonic ectoderm (ExEct) ... . Such cells are ---“, please see lines 185–186 in the revised text.
For line 187, containing the phrase “... method to assess which tools ...”, please see lines 196–197 in the revised text.
For line 189, containing the phrase “... serious damage to PGCs ...”, please see line 198 in the revised text.
For line 195, “... genital ridges are located on the posterior aspect of the embryo ...”, please see lines 204–205 in the revised text.
For line 195, containing the phrase “Svingen et al. developed ...”, please see line 215 in the revised text.
For line 207, containing the phrase "… injected into isolated ...", please see line 217 in the revised text.
For line 214, the previously used phrase was deleted (please see line 225 in the revised text).
- 2.4 Gene Delivery to SSCs. Line 232: "that are immature and capable of self-renewal, but maintain the ability to differentiate..."Lines 234-235: "which is called" not necessary. Just put (male germ-cell transplantation"). Line 253: "stimulated by stimulated by retinoid acid 8..": rephrase.
Answer: As suggested, we have corrected these lines as follows:
For line 232, "… that are immature and capable of self-renewal, but maintain the ability to differentiate ...", please see lines 251–252 in the revised text.
For lines 234–235, containing the phrase “... (male germ-cell transplantation) [63].”, please see lines 254 in the revised text.
For line 253, containing the phrase "... activated by stimulated by retinoid acid 8, ...", please see line 273–274 in the revised text.
- 2.5. Infection of preimplantation .... Lines 265-266: The zona pellucida is a layer of glycoproteins with several important functions. the statement that is a glycoprotein protecting from hazardous substances is not correct. Line 268: " "(the space between the zona pellucida and the membrane of the oocyte)". "in a medium...": "in a drop of medium containing...?". line 279: "to the the tyrosinase (Tyr) gene coding". Line 285: rephrase the sentence.
Answer: As suggested, we have made the following corrections:
For line 265–266, "ZP is a layer of glycoproteins with several important functions ...“, please see line 286 in the revised text.
For line 268, “... (the space between the ZP and the membrane of the oocyte) or in a drop of medium containing ...“, please see lines 289–290 in the revised text.
For line 279, containing the phrase "... to the tyrosinase (Tyr) gene coding ...", please see line 301 in the revised text.
For line 285, the sentence used previously was deleted. We have added the following sentence: “Mizuno et al. [9] observed that among the rAAVs that were tested, ...”. Please see line 307 in the revised text.
- 2.6. Gene delivery to post-implantation....Line 308: "in blocking immunological attacks by maternal immune cells"; Line 309: "support to the embryo"; line 311: "embryo visualization"; "Temporal: correct to Temporary; Line 330: "on successful in vitro genome..."
Answer: As suggested, we have corrected these lines as follows:
For line 308, "in blocking immunological attacks by maternal immune cells ...", please see lines 332–333 in the revised text.
For line 309, containing the phrase "support to the embryo", please see line 333 in the revised text.
For line 311, containing the phrase "embryo visualization", please see line 335 in the revised text.
For line 311, containing the phrase “Temporary in vitro ...”, please see line 335 in the revised text.
For line 330, containing the phrase "on successful in vitro genome ...", please see line 354 in the revised text.
- 4. Concluding remarks: The last paragraph (lines 483-493) should be brought up at the beginning, integrating the concepts with those expressed in lines 458-459. The clinical applications here lack of a reference and are in contrast with the concepts expressed below. Line 463: "each of these technologies has specific characteristics." Line 467-468: "than the above mentioned techniques " is not necessary;
Answer: In the concluding remarks we have made a significant change, based on the reviewer’s suggestion. Please see lines 486–512 in the revised text.
- Table 1: Title: Summary of the characteristic properties of the ....editing techniques... Column headings: "Delivery mode(s)"; "Method(s)", "Equipment and others???" What do you mean with "others"?; . Please reduce the text in the columns to the essential.
Answer: As suggested, we have improved the content in Table 1 in the revised text.
- Figure 2: Please rewrite the legend. What does "Table 2" refer to?
Answer: As suggested, we have rewritten the legend of Figure 2 in the revised text. We have also re-made Figure 2 itself.
- Figure 3, legend: "Schematic representation of..." line 263: "to generate KI pups".
Answer: As suggested, we have improved this portion in the revised text (Please see lines 278 and 284).
- Figure 4: Provide a better annotation for day 7.5 embryonic stage. "Drop containing plasmid DNA" Legend: schematic representation of....
Answer: As suggested, we have rewritten the legend of Figure 4 in the revised text (Please see lines 323 and 327–284).
- Figure 5, legend, line 367: "through the oviductal..."
Answer: As suggested, we have rewritten the legend of Figure 5 in the revised text (Please see line 391).
- Figure 6, Legend: on day = at day; line 390. "is transferred to the embryo via the vitelline..." Line 393: "MDPI, publisher of...."
Answer: As suggested, we have rewritten the legend of Figure 6 in the revised text (Please see lines 417 and 420).
